# The Effect of Lower Body Anaerobic Pre-loading on Upper Body Ergometer Time Trial Performance

**DOI:** 10.3390/sports9060079

**Published:** 2021-05-31

**Authors:** Priit Purge, Dmitri Valiulin, Allar Kivil, Alexander Müller, Gerhard Tschakert, Jaak Jürimäe, Peter Hofmann

**Affiliations:** 1Institute of Sport Sciences and Physiotherapy, Faculty of Medicine, University of Tartu, 50090 Tartu, Estonia; priit.purge@ut.ee (P.P.); dmitri.valiulin@gmail.com (D.V.); allar.kivil@ut.ee (A.K.); jaak.jurimae@ut.ee (J.J.); 2Training & Training Therapy Research Group, Institute of Human Movement Science, Sport & Health, Exercise Physiology, University of Graz, 8010 Graz, Austria; alexander.mueller@uni-graz.at (A.M.); gerhard.tschakert@uni-graz.at (G.T.)

**Keywords:** cross-country, performance, pre-load, lactate, upper body, glycolysis

## Abstract

Pre-competitive conditioning has become a substantial part of successful performance. In addition to temperature changes, a metabolic conditioning can have a significant effect on the outcome, although the right dosage of such a method remains unclear. The main goal of the investigation was to measure how a lower body high-intensity anaerobic cycling pre-load exercise (HIE) of 25 s affects cardiorespiratory and metabolic responses in subsequent upper body performance. Thirteen well-trained college-level male cross-country skiers (18.1 ± 2.9 years; 70.8 ± 7.6 kg; 180.6 ± 4.7 cm; 15.5 ± 3.5% body fat) participated in the study. The athletes performed a 1000-m maximal double-poling upper body ergometer time trial performance test (TT) twice. One TT was preceded by a conventional low intensity warm-up (TT_low_) while additional HIE cycling was performed 9 min before the other TT (TT_high_). Maximal double-poling performance after the TT_low_ (225.1 ± 17.6 s) was similar (*p* > 0.05) to the TT_high_ (226.1 ± 15.7 s). Net blood lactate (La) increase (delta from end of TT minus start) from the start to the end of the TT_low_ was 10.5 ± 2.2 mmol L^−1^ and 6.5 ± 3.4 mmol L^−1^ in TT_high_ (*p* < 0.05). La net changes during recovery were similar for both protocols, remaining 13.5% higher in TT_high_ group even 6 min after the maximal test. VCO_2_ was lower (*p* < 0.05) during the last 400-m split in TT_high_, however during the other splits no differences were found (*p* < 0.05). Respiratory exchange ratio (RER) was significantly lower in TT_high_ in the third, fourth and the fifth 200 m split. Participants individual pacing strategies showed high relation (*p* < 0.05) between slower start and faster performance. In conclusion, anaerobic metabolic pre-conditioning leg exercise significantly reduced net-La increase, but all-out upper body performance was similar in both conditions. The pre-conditioning method may have some potential but needs to be combined with a pacing strategy different from the usual warm-up procedure.

## 1. Introduction

Pre-loading or warm-up are common practices before training or competition. Traditional warm-up effects have been proved beyond doubt for achieving better preparation and readiness for subsequent performance. The scientific community supports the use of warm-up, which has been reported to increase muscle temperature and stimulate performance of muscle contraction [1]. Increased body temperature and speeded VO_2_ kinetics are now on top of the list of gaining effects. Since athletic skills and physiological properties have increased rapidly in the late years, the focus has moved to better physiological preparation [2,3], that could speed up VO_2_ kinetics and increase peak power output (PPO) values without spending crucial energy and time. According to this approach, Müller et al. [4] found that hand crank exercise before leg exercise significantly influenced anaerobic energy contribution during the leg workout. Several authors have recently shown the effects of the contrary limbs priming effect which may be a new preparation method, prescribed as metabolic conditioning [5,6]. The main idea is that elevated lactate concentration in blood induces an inward gradient directed to lactate uptake after exercise and in the case of a subsequent high-intensity exercise inhibits lactate export from working muscle thus inhibiting lactate production and possibly enhancing oxidative metabolism [4,7].

Therefore, using contrary limbs for a high-intensity warm-up or pre-loading it is possible to increase the systemic blood lactate (La) level before the main exercise bout priming the muscles with a negative lactate gradient such as lactate uptake rather than lactate export [6]. This interaction between La producing and La consuming muscle cells and their interaction with the body system can be explained by the lactate shuttle theory [8,9]. It is suggested that short-term performance which is dependent on the ability to break down high-energy phosphate stores, will be negatively affected by an intense warm up that decreases the availability of high-energy phosphates [10]. On the other hand, exercise longer than 2–3 min is primarily limited by aerobic energy contribution which may be enhanced by intense warm-up procedures [11].

One purpose of warm-up is to prepare the body for high-intensity work and to speed up aerobic energy production [5]. For performances that require (sub)maximal aerobic metabolism, conditioning can be achieved both by moderate and heavy exercises equally [12]. Active warm-up is crucial before high-intensity exercise, being responsible for physiological changes the combined effect of which is capable of improving performance [10]. Main effects are perhaps increased muscle temperature, nerve conduction rate, speeded metabolism kinetics, oxygen uptake kinetics, muscle post-activation potentiation, and psychological preparedness [10,13]. Although excessively intense exercise before a high-intensity bout may reduce performance level through decreased muscle glycogenolysis and La production [14,15], lower levels of muscle glycogen and pH, the effects of anaerobic pre-load with non-dominant muscles remains unclear.

In addition, elevated blood La concentration per se could accelerate oxygen uptake (VO_2_) at the start of exercise by forcing the organism to stimulate mitochondrial respiration [16]. This effect is suggested from an inhibition of lactate production due to an inverse gradient for lactate due to already elevated blood lactate concentration from non-dominant muscles [7]. Purge et al. [5] found that pre-loaded high-intensity exercise conducted by the upper body affected glycogenolysis and significantly reduced carbon dioxide output (VCO_2_) during 7–8 min of maximal performance in rowers. Systemic blood La concentration can be easily increased by few muscles [5], which could be useful if they are not dominant during a subsequent maximal performance [4]. In line with these studies are the results from Birnbaumer et al. [6] who showed that anaerobic metabolic pre-conditioning significantly improved pull-up exercise compared to a standard warm-up program. It has been substantiated previously that blood La concentration is related to VCO_2_ [17] and thus to anaerobic effort [10,18,19]. Gaesser et al. [20] reported that infusions of adrenaline markedly elevated blood La levels without altering VO_2_ during heavy exercise. In addition, Bohnert et al. [21] showed how to increase VO_2_ kinetics during subsequent heavy lower body exercise, with previous high-intensity arm crank exercise. Although the magnitude was lower, but qualitatively comparable, it needs to be mentioned that some of these findings may be more coincidental rather than a cause and effect.

In recent studies it has been found that a previous rise of blood La level decreased La production in a following maximal performance trial [22]. Similar results were found in studies of Müller et al. [4], Purge et al. [5] and Birnbaumer et al. [6], where performance level could be persisted or improved. In Müller et al. [4] and Purge et al. [5] the priming action was executed with arm crank exercise and TT with lower body, whereas Birnbaumer et al. [6] used the opposite strategy. Thus, priming effects can be reached by a short high-intensity exercise bout applying non-specific muscle groups and so speeding up VO_2_ kinetics highly important for a prolonged race duration [2,11]. A similar effect has been already shown for the ingestion induced acidosis by Brien and McKenzie [23] in rowers. A major problem with high-intensity, long-duration exercise is tolerating the side effects of the highly anaerobic first 40–60 s of exercise. Therefore, it was suggested that an early accelerated increase in VO_2_ and a lower La production within the working muscle improves performance [24]. Recently, Purge et al. [5] presented, that a high intensity upper body pre-load increased the systemic blood La concentration to 8–9 mmol L^−1^, while reducing the net blood La increase during maximal performance to 53.3% compared to a usual warm-up without upper body high intensity pre-load. However, no significant change in performance time for 2000 m ergometer rowing distance was found [5].

Although physiological benefits can be suggested from previous studies, there is still some need regarding the optimal dosage of such a conditioning protocol before integrating such priming activities into competitive sports. The main aim of this study was, therefore, to investigate the physiological and performance effects of high-intensity cycling leg exercise on subsequent upper-body double poling exercise performance.

## 2. Materials and Methods

### 2.1. Participants

The study sample consisted of 13 well-trained college-level male cross-country skiers (age: 18.3 ± 2.9 years.; body mass: 70.8 ± 7.3 kg; body height: 180.8 ± 4.6 cm; body fat: 15.5 ± 3.5%). Data collection was scheduled between off-season preparations and first outdoor competition in spring 2017. Inclusion criteria were that all participants were experienced national level athletes, being familiar with indoor skiing competition procedures on C2 ski ergometer (SkiErg, Concept2, Inc., Morrisville, VT, USA, 2009^®^) and have been training for at least 5 years. Exclusion criteria were illness or injury. The participants were introduced to experimental methods and were asked to sign written consent to participate in the study approved by the Ethics Committee of the local University in line with ethics of the World Medical Association (Declaration of Helsinki, 7th edition) [25].

### 2.2. General Procedure

Participants attended the laboratory on three occasions within a 3-week period, with each visit being separated by at least 3 recovery days. On the first visit participants’ body composition was measured using DXA (Hologic Discovery DXA, Marlborough, MA, USA) [26] and they performed a specialist supervised incremental double-pull exercise on an upper body Skiing ergometer [27,28]. A skiing ergometer is the normal training ergometer for skiers in summertime. On two subsequent occasions time trial performance tests (TT) were performed [5]. One with upper body preload and one without. Gas exchange variables, heart rate and blood lactate concentration (La) were measured in all tests.

### 2.3. Body Composition

Anthropometric parameters were measured with precision of 0.1 cm of height (Martin metal anthropometer) and 0.05 kg of body mass (A&D Instruments, Abingdon, UK). For body composition assessment, a DXA scanner (Hologic, Marlborough, MA, USA) was used. Body fat mass (FM) and lean body mass (LBM) were evaluated using a dual-energy X-ray absorptiometry (DXA).

### 2.4. Incremental Double Pull Exercise on Upper Body Skiing Ergometer

During the first visit all participants undertook an incremental test with upper body double poling on a ski ergometer (Concept2, Morrisville, VT, USA) [27], to determine the respiratory exchange parameters such as maximal (VO_2max_; L min^−1^) and relative (VO_2max_/kg; mL min^−1^ kg^−1^) oxygen consumption, minute ventilation (VE; L min^−1^), respiratory exchange ratio (RER), maximal aerobic power (P_max_; W), as well as first (VT_1_) and second (VT_2_) ventilatory thresholds (Cortex MetaMax 3B, Leipzig, Germany) [29]. The test started with an initial work rate of 40 W with increments of 20 W after every min until volitional exhaustion [5]. The analyser was calibrated before the test with barometric pressure, ambient air, humidity readings and gases of known concentrations. Heart rate (HR) was recorded using a chest strap and HR monitor (Polar RS 800, Polar Electro, Kempele, Finland) every 5 s during the test. The thresholds (VT_1_ and VT_2_) were determined considering the first rise in ventilation between the first workload and 60% of P_max_ (VT_1_) and the second abrupt rise in ventilation between VT_1_ and P_max_ (VT_2_) [29]. Calculations of turn points were performed by means of linear regression turn point analysis [29] and adjusted by an experienced specialist if needed. All data were analysed by means of computer support using standard software (MetaMax-Analysis 3.21, Cortex, Leipzig, Germany). The athletes were fully familiarized with the use of this C2 apparatus. Distance, power and stroke frequency were delivered continuously on the computer display of the C2 ski ergometer and were collected simultaneously and averaged on a stroke-by-stroke basis.

### 2.5. Skiing Ergometer Performance

Experimental all-out tests in an indoor environment were maximally converged to competitive situations using a wind-resistance-braked skiing ergometer (SkiErg, Concept 2, Morrisville, VT, USA). This mode of exercise is conducted by repeatedly pulling down the handles from above the shoulders that are connected to a pulley system. It is characterized by the work of mainly the upper body [27].

A distance of 1000 m was covered in the shortest time possible. Athletes were allowed to choose ergometer resistance level between 4 and 6 but were not allowed to check for stroke rate and pace during the trial. Power and stroke frequency were recorded continuously to analyse pace strategy and speed for different parts of distance.

Two protocols were performed to measure the effect of a high-intensity anaerobic warm-up phase on all-out cross-country ski endurance performance. Both protocols were aiming to measure parameters of TT which in one case was following a low-intensity warm-up (TT_low_) with a 14 min duration of recovery between warm-up and workout. In the second protocol (TT_high_), an additional 25 s high-intensity anaerobic cycling pre-load protocol was performed 5 min after the end of warm up, leaving approximately 9 min for recovery before the 1000 m all-out skiing ergometer TT (Figure 1). The order of tests was randomized for each participant to minimize learning effects and exclude a possible advantage of one or other protocol sequence. The warm-up in TT_low_ consisted of a 20 min workload at 50% VO_2max_ from the incremental exercise test. In the experimental setting a 25 s high-intensity anaerobic lower body cycling pre-load exercise (HIE) was added to cause a significant increase in blood La concentration.

HIE was performed on a mechanically braked stationary cycle ergometer (Monark Ergomedic 849E, Varberg, Sweden) applying a brake weight individually of 70 g kg^−1^ body weight. Respiratory performance and gas exchange during TT’s were measured using a face mask (Cortex MetaMax 3B, Leipzig, Germany) with online tracking of respiratory fatigue. Participants had to have slept well, had sufficient nutrition and be hydrated before performing each protocol; at least 3 days off were agreed to precede a 1000 m TT. Blood La concentration was determined for both conditions at rest, after the warm-up, after the anaerobic-preload, and during recovery (Figure 1). Blood La as well as blood glucose concentrations were measured by means of capillary samples (20 µL) from a hyperemised earlobe (Biosen S-line, EKF-Diagnostic, Germany). Pre- and Post-exercise rate of perceived exertion (RPE) was assessed using the Borg CR-10 scale [30].

### 2.6. Statistical Analysis

Data are expressed as mean ± SD. All variables were first checked for normality using the Shapiro–Wilks method. To assess the difference of variables between the two TT tests, data were analysed using a two-way repeated measures analysis of variance (ANOVA) and post hoc test on both factors (test and time). Paired sample t-tests were used to assess differences between the variables. Effect size statistics (ES) for selected variables was calculated. These calculations were based on Cohen’s classification for a small (0.2 ≤ d < 0.5), moderate (0.5 ≤ d <0.8), and large (d ≥ 0.8) ES [31]. To assess the relationship between 1000 m performance results variables and relevant gas-exchange and La concentrations, Pearson correlation analysis was used. The magnitude of the correlation coefficients was determined as trivial (r < 0.1), small (0.1 < r < 0.3), moderate (0.3 < r < 0.5), high (0.5 < r < 0.7), very high (0.7 < r < 0.9), nearly perfect (r > 0.9), and perfect (r = 1) [32]. All statistical procedures were calculated using the statistical package SPSS (version 20) (IBM, Armonk, NY, USA). Statistical significance was set at *p* < 0.05.

## 3. Results

All 13 participants completed the incremental skiing ergometer test and both two TT tests in a randomized order. All performance variables from the incremental double pulling exercise test are presented in Table 1. In brief, VO_2max_ was found at 57.32 ± 5.30 mL kg^−1^ min^−1^ and P_max_ was 270.1 ± 42.1 W.

Ski ergometer maximal performance during TT_low_ was not significantly faster compared to the TT_high_ condition (225.1 ± 17.6 s vs. 226.1 ± 15.7 s; *p* > 0.05). In TT_low_ the participants were tended to be slower in the first 400 m, but tended to be faster in the last 600 m (Table 2).

VO_2_ was significantly higher in TT_high_ (*p* < 0.05) in the first 400 m split, however, in the other splits we found no significant differences between the two protocols. VCO_2_ increase was significantly lower at the start (*p* < 0.05) and in the fourth and fifth 200 m split in the TT_high_ performance test. Due to the pre-load exercise, RER in TT_high_ was significantly higher at the start but decreased during the TT and was significantly lower compared to TT_low_ in the second, third, fourth and fifth 200 m split. Participants presented a higher heart rate at rest and 200 m (*p* < 0.05); however, HR was not significantly different from 400 m to the end of the TT (Figure 2).

Blood La concentrations after low-intensity warm-up were not significantly different between both test protocols (Table 3). The 25 s high intensity anaerobic cycle ergometer exercise significantly increased blood La concentration (*p* < 0.05) before the TT in TT_high_ (La_before_: 8.2 ± 2.2 mmol L^−1^) compared to TT_low_ (1.4 ± 0.3 mmol L^−1^). Additionally, blood La concentration after the TT (La_after_) was significantly higher in TT_high_ (14.7 ± 4.6 mmol L^−1^) compared to TT_low_ (11.8 ± 2.4 mmol L^−1^) but glucose concentration was not significantly different (Table 3). However, net La increase in TT_low_ was 10.5 ± 2.2 mmol L^−1^ (Figure 3) which was significantly higher (*p* < 0.05) compared to TT_high_ in which net La increase was reduced by approximately 50%. Net La changes during recovery were similar for both protocols; however, remaining 13.5% higher in TT_high_ group even 15 min after the maximal test. Additionally, rating of perceived exertion before tests (RPE_Pre_) was significantly (*p* < 0.05) lower in TT_low_ compared to TT_high_ and also after all-out exercise (Table 2).

We found a significant and high relation (R^2^ = 0.410, *p* < 0.05) between the first 200 m split speed and in total finish time in TT_low_ and TT_high_ tests. Figure 4 shows that starting the first 200 m split slower following TT_high_ compared to TT_low_ had faster total times than in TT_low_, while subjects that did not adapt their pacing following TT_high_ or even went out slightly faster in the first 200 m showed a reduction of performance.

## 4. Discussion

This study aimed to measure the effects of 25 s of lower body anaerobic HIE on subsequent upper body performance. The main finding of the present study was that 4 min maximal upper body performance time was not significantly influenced by high-intensity lower body cycling pre-load exercise although anaerobic energy contribution of the arms indicated by lower net lactate increase, lower VCO_2_ and RER, was substantially reduced (Table 2). Oxygen uptake was significantly increased for the first 400 m of TT_high_, possibly caused by an increased ventilation in the TT_high_ condition such as a priming effect [12].

The four-minute upper-body maximal performance is highly related to VO_2max_ and locomotion’s oxygen cost [33]. The distribution of aerobic versus anaerobic energy supply was prescribed to be ~70%/30% [33], which is similar to what was found for other sports of the same duration [34]. It is a fact that anaerobic energy contribution is important for successful maximal performance, as it can fulfil fast and explosive energy demands [35]. Energy system contribution in cross-country skiing was shown to be similar to running, cycling and rowing [33] at least with the applied duration of maximal exercise in our study. On the other hand, HIE preload as applied in our study reduced lactate increase within the working muscle without reducing overall performance.

Importantly, the effect of a high-intensity anaerobic start needs to be elucidated. High-intensity effort increases muscle and blood La concentrations and decreases pH already in the first min of a 3–4 min race if starting La concentration is low [24]. It is well known that anaerobic glycolysis is at its highest after about 40–50 s and starts to decrease thereafter due to inhibitory effects of the high La, low pH situation [36]. It was shown that pre-elevation of systemic La levels by non-specific muscle exercise inhibited subsequent net La production, an effect which is suggested to be applied in competitive high-intensity exercise to improve performance [4]. Two main effects may be expected, such as a decrease of anaerobic energy production during the first minute of a race and a resulting increased oxidative energy contribution [5,6]. Developing this idea opens new doors understanding the metabolic background of competitions longer than 2–3 min. On the one hand the strategy of metabolic pre-conditioning could be helpful in the prologue but on the other hand the higher La levels during recovery may be disadvantageous for the subsequent exercise bouts. Our study did not show an overall performance enhancing effect, although some details suggest beneficial effects not realized due to an unfavourable pacing strategy. Figure 4 shows that starting the first 200 m split faster than in TT_low_ resulted in a prolonged total time compared to the TT_low_ group which may be explained by the fact that a lower anaerobic contribution suggested participants start faster. Therefore, according to our study results, refraining from a faster start is therefore necessary with respect to overall performance time, which could allow athletes to gain the effects of a reduced anaerobic energy contribution during the first minute of high-intensity anaerobic exercise. We may argue that forcing an athlete’s metabolism towards aerobic metabolism by inhibiting anaerobic glycolysis during the first minute of exercise can preserve muscles’ abilities to perform on a high level for finishing the distance without losing overall performance, although the optimal dosage of pre-load and recovery remains unclear and needs further investigations. Pacing the workload was shown to increase overall performance in the pull-up exercise [6], which indicates that a different pacing strategy needs to be developed applying such as metabolic pre-conditioning warm-up compared to the usual pacing strategy.

Anderson et al. [37] found that 18% of cross-country skiers consumed almost all of their accumulated oxygen (O_2_) and total anaerobic resources over the first 550 m, while O_2_ deficit was decreased during the final 300 m. In our investigation, athletes reached their VO_2_ maximum at the end of the first 400 m split (Figure 3) and it was similar in both TT_high_ and TT_low_ tests. In the TT_high_ test the athletes performed better during the first 400 m, but TT_low_ showed faster times during the last 400 m. Both results present a non-significant difference (*p* > 0.05). The decrease in overall performance in TT_high_ compared to TT_low_ was related to a small number of participants starting too fast (Figure 4) but the others were able to perform faster although a reduced anaerobic contribution was indicated by a significant decrease in net La concentration and in CO_2_ (Figure 3). Maximal blood La concentrations in our study were found at 11.8 ± 2.4 mmol L^−1^ after 1000 m in TT_low_, and 14.7 ± 4.6 mmol L^−1^ in TT_high_ which is comparable to the results of our previous studies in rowers [5]. The metabolic pre-conditioning elevated both pre and post all-out exercise La concentration but was well within tolerable limits.

Our results are consistent with other studies which found that sprint pre-exercise, elevating baseline blood La up to ~6 mmol L^−1^, did not significantly diminish performance [16]. In our study, blood La before the start of TT was elevated to 8.2 ± 2.2 mmol L^−1^ by the pre-conditioning load, comparable to the study by Birnbaumer et al. [6].

Sprint exercise, which induced a lasting lactic acidosis, before a heavy exercise bout may be compared to the applied HIE in our study as the physiological influence on VO_2_ response was similar [16]. This implies that high-intensity anaerobic pre-load exercise without sufficient recovery may limit athletic performance and maximal effort capacity, as long as maximal anaerobic energy contribution is required [7,17,38]. Burnley et al. [2] reported that 10 min recovery from previous high-intensity loading may be insufficient to provide a significant priming effect. Our results neither showed a significant priming effect nor a significant increase in VO_2_ during peak performance questioning these effects although an increased oxygen uptake has been reported in a similar study with specific pull-up exercise [6]. However, in our study overall performance was similar in both interventions although a clear decrease of anaerobic energy contribution. Slower TT_high_ split times were only significant for the last 400 m, which was influenced by the starting speed but not the pre-load situation (Figure 2). It is suggested that adequate pacing strategies are necessary if applying such a metabolic pre-conditioning method. In support of our hypothesis the results show a significant reduction of the increase in net lactate concentration without significant differences for overall peak and mean power which is consistent with Bogdanis et al. [22] and Müller et al. [4]. However, this intervention may be beneficial as in some cases even an increased performance was found [6] if La elevation was induced by muscle groups not involved in the main exercise workout. Within the investigated La elevation, pre-exercise blood La concentration and TT performance in TT_high_ were not significantly related although the optimal limits of La increase need to be elucidated.

As a limit it needs to be mentioned that TT_high_ poor performance in some participants in our study could be caused because athletes were not frequently using the skiing ergometer, as many of them prefer to use roller skis as an alternative for summer season. In the absence of experience of using it for maximal performance, starting and finishing pace strategies were based simply on the first laboratory test. Additionally, our experiment is limited to the influence of high-intensity anaerobic metabolic preconditioning exercise on just a single subsequent bout of cross-country specific ergometer exercise but did not focus on several repeated exercise bouts such as in a real-life competition [39]. As the 25 s maximal priming exercise with the lower body followed by 9 min of rest neither had a positive nor negative overall effect on upper body short lasting time trial performance in our study, there may be other combinations of priming exercise duration and rest that may enhance performance.

However, the present study may have some practical applications. First, anaerobic HIE performed with non-dominant muscles with respect to the main task has the potential to inhibit the net lactate increase at the start of exercise, and when accompanied by an appropriate pacing strategy may improve performance. Second, elevated systemic La levels do not significantly impair subsequent peak and mean power, when provoked by muscle groups not involved in the main exercise workout.

## 5. Conclusions

HIE anaerobic pre-conditioning exercise when added to a standard low-intensity warm-up protocol failed to improve overall short-term double poling performance but opened some questions regarding the optimal preparation before a high-intensity all-out workout as well as the adequate pacing strategy. Additional studies are needed before any conclusions can be drawn. Future studies should focus on different HIE pre-load strategies, different rest periods before maximal performance test and specified pacing during the race.

## Figures and Tables

**Figure 1 sports-09-00079-f001:**
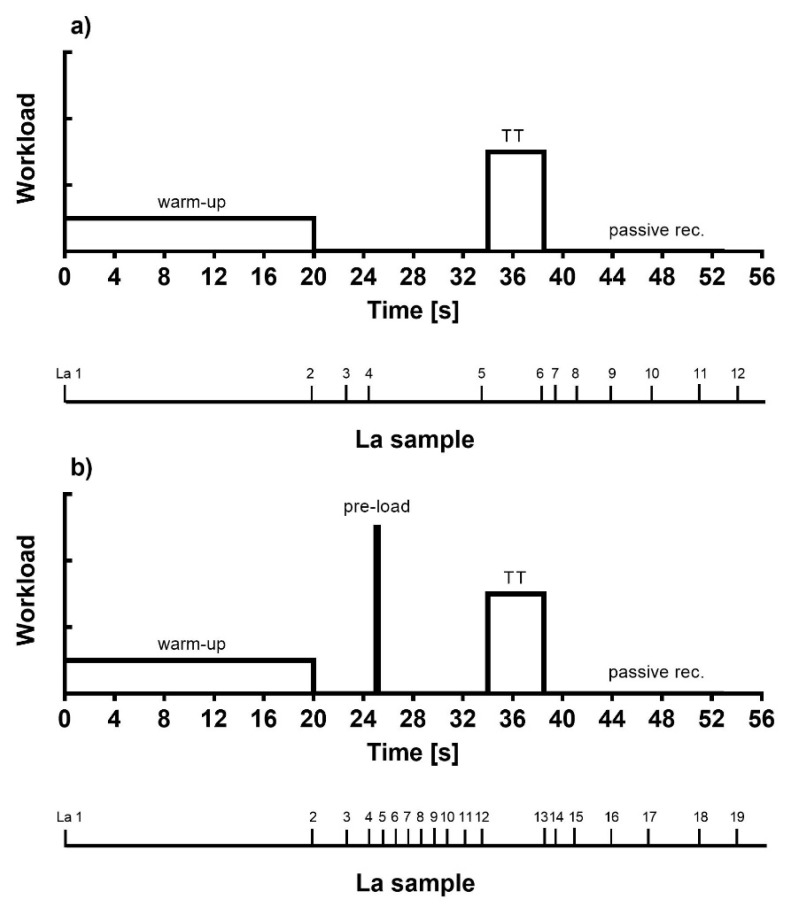
Protocols for pre-load influence without pre-load (**a**) and with pre-load (**b**) on 1000 m performance cross-country sprint ergometer exercise testing. High-intensity exercise (pre-load) indicated the high intensity 25 s all-out cycling pre-load to induce an elevated blood lactate concentration before the maximal 1000 m all-out SkiErg time trial performance tests (TT). Numbers in the upper row indicate time in min (t_0–56_) and numbers in the lower row (La_1–19_) indicate the number of blood samples to measure blood lactate concentration (La).

**Figure 2 sports-09-00079-f002:**
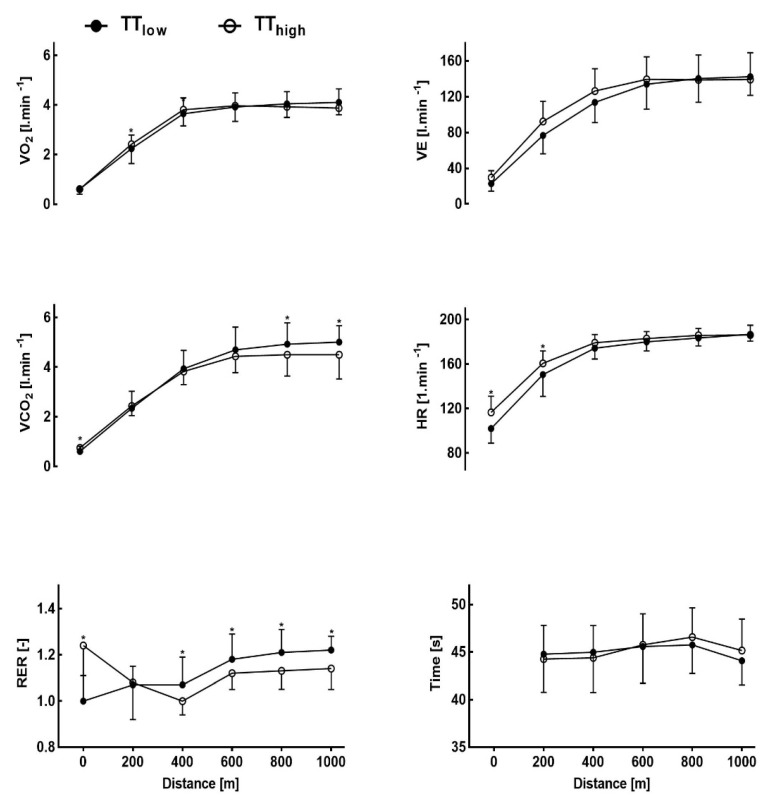
Oxygen uptake (VO_2_), carbon dioxide release (VCO_2_) and respiratory exchange ratio (RER) (left) as well as Ventilation (VE), heart rate (HR) and 200 m split times (right) after usual low intensity warm up (TT_low_) and additional 25 s all-out cycling pre-load (TT_high_) in a 1000 m all-out upper body Ski-ergometer time trial test (TT). * Significantly different from TT_low_ (*p* < 0.05)

**Figure 3 sports-09-00079-f003:**
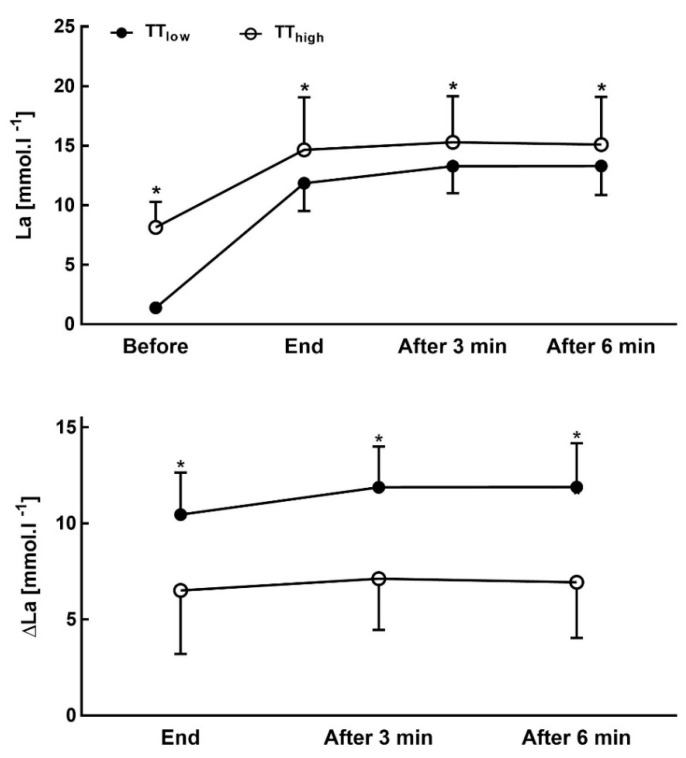
Absolute (above) and net (below) blood lactate concentration at the start, at the end and after 3, and 6 min of recovery in a maximal 1000 m all-out Ski-ergometer time trial (TT) performance test with low intensity (TT_low_) and with 25 s all-out cycling pre-load exercise (TT_high_). * Significantly different from TT_low_ (*p* < 0.05)

**Figure 4 sports-09-00079-f004:**
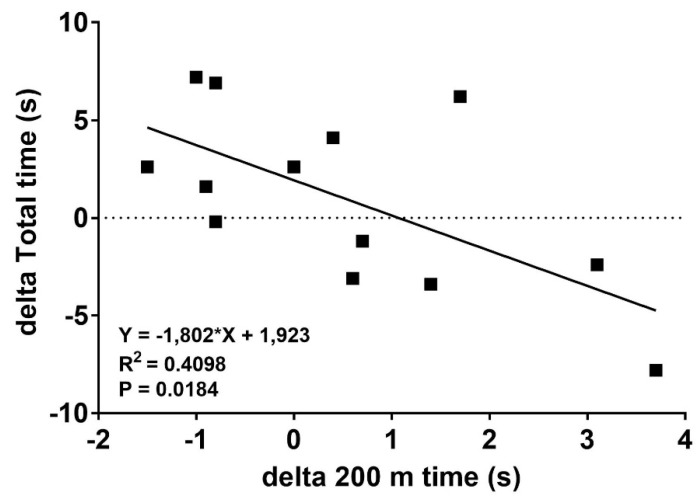
The relationship between first 200 m split time (difference between TT_low_ and TT_high_) and final difference in overall time in a maximal 1000 m all-out Ski-ergometer time trial (TT) performance test with low intensity (TT_low_) and with 25 s all-out cycling pre-load exercise (TT_high_).

**Table 1 sports-09-00079-t001:** Body composition and test characteristics from the maximal incremental Ski ergometer exercise test.

Participants (*n* = 13)	Min	Max	Mean ± SD
Age (years)	15.4	24.9	18.3 ± 2.9
Height (cm)	171.0	190.0	180.8 ± 4.6
Body Mass (kg)	55.2	82.7	70.8 ± 7.3
Fat (%)	12.9	26.3	15.5 ± 3.5
VO_2max_ (L/min^−1^)	2.85	5.19	4.06 ± 0.58
VO_2max_ (mL min^−1^ kg^−1^)	49.82	66.42	57.32 ± 5.30
VE_max_ (L min^−1^)	109.90	184.60	146.6 ± 24.3
HR at VT_1_ (beats min^−1^)	136	178	154.2 ± 11.4
P at VT_1_ (W)	82	163	127.0 ± 25.5
HR at VT_2_ (beats min^−1^)	165	195	181.3 ± 8.5
P at VT_2_ (W)	129	222	186.2 ± 28.2
P_max_ (W)	180	345	270.1 ± 42.1
HR_max_ (beats min^−1^)	185	206	198.6 ± 6.9

Maximal oxygen uptake (VO_2max_), maximal ventilation (VE_max_), heart rate (HR), first ventilatory threshold (VT_1_), second ventilatory threshold (VT_2_), power (P), maximal power (P_max_), maximal heart rate (HR_max_).

**Table 2 sports-09-00079-t002:** The results of maximal 1000 m all-out Ski ergometer performance test (TT) with low intensity warm up (TT_low_) and additional high intensity 25 s all-out cycling pre-load (TT_high_).

Participants (*n* = 13)	Time_200 m_ (s)	Time_400 m_ (s)	Time_600 m_ (s)	Time_800 m_ (s)	Time_1000 m_ (s)	Time (s)	RPE_before_	RPE_after_
TT_low_	44.8 ± 4.2	44.9 ± 4.4	45.6 ± 4.0	45.7 ± 3.1	44.1 ± 2.6	225.1 ± 17.6	0.6 ± 0.8	8.9 ± 0.9
TT_high_	44.3 ± 3.7	44.4 ± 3.5	45.8 ± 3.4	46.6 ± 3.2	45.2 ± 3.5	226.1 ± 15.7	1.4 ± 1.2	9.5 ± 0.7
Dif	0.5 ± 1.6	0.6 ± 1.6	−0.2 ± 1.4	−0.8 ± 1.5	−1.1 ± 1.9	−0.9 ± 4.6	−0.8 ± 1.2	−0.7 ± 1.1
p	0.28	0.21	0.05 *	0.07	0.07	0.47	0.04 *	0.05 *
Effect size	0.12	0.11	0.05	0.29	0.42	0.06	1.0	0.67

* Significantly different from TT_low_ (*p* < 0.05); Effect size: very small (0.01–0.20), small (0.20–0.50), medium (0.50–0.80), large (0.8–1.2), very large (1.2–2.0), huge (>2.0).

**Table 3 sports-09-00079-t003:** Blood lactate and glucose concentration (mmol L^−1^) before and after low intensity warm up, and before and after 3, 6, 9, 12 and 15 min of maximal 1000 m SkiErg performance test.

N = 13		Before Warm-Up	After Warm-Up	Before TT	After TT	After TT 3 min	After TT 6 min	After TT 9 min	After TT 12 min	After TT 15 min
Lactate	TT_low_	1.9 ± 0.7	2.0 ± 0.5	1.4 ± 0.3	11.8 ± 2.4	13.3 ± 2.4	13.3 ± 2.4	12.7 ± 2.7	12.1 ± 2.6	10.9 ± 2.9
	TT_high_	1.8 ± 0.6	2.0 ± 0.5	8.2 ± 2.2 *	14.7 ± 4.6 *	15.3 ± 4.0 *	15.1 ± 4.2 *	14.7 ± 4.1 *	13.7 ± 4.1 *	12.6 ± 4.3 *
	Difference	5.6%	0%	82.9%	19.7%	13.1%	11.9%	13.6%	11.7%	13.5%
	Effect Size	0.2	0.0	4.3 *	0.8	0.6	0.5	0.6	0.5	0.5
Glucose	TT_low_	5.3 ± 0.9	4.5 ± 0.5	4.9 ± 0.5	5.0 ± 0.4	6.2 ± 0.7	6.0 ± 0.8	5.7 ± 0.6	5.6 ± 0.6	5.5 ± 0.6
	TT_high_	5.3 ± 1.0	4.5 ± 0.4	4.5 ± 0.8	4.9 ± 1.0	6.2 ± 1.0	5.9 ± 1.1	5.8 ± 0.8	5.6 ± 0.7	5.4 ± 0.9
	Difference	0.0%	0.0%	8.2%	2.0%	0.0%	1.7%	1.7%	0.0%	1.8%
	Effect Size	0.0	0.0	0.6	0.1	0.0	0.1	0.1	0.0	0.1

TT_low_- Maximal 1000 m SkiErg performance test with normal low intensity warm-up; TT_high_- maximal 1000 m SkiErg performance test with additional 25 s all-out cycling pre-load. * Significantly different from TT_low_ (*p* < 0.05); Effect size: very small (0.01–0.20), small (0.20–0.50), medium (0.50–0.80), Large (0.8–1.2), very large (1.2–2.0), huge (>2.0).

## Data Availability

The data presented in this study are available on a request from the corresponding author for researchers who meet the criteria for access to confidential data.

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
