# Peer review of "The Effect of Lower Body Anaerobic Pre-loading on Upper Body Ergometer Time Trial Performance"

_sports, 2021, doi:10.3390/sports9060079_

Round 1
Reviewer 1 Report
Dear authors:
After review the document. Here some ideas in order to improve the final text.
King Regards

Author Response
Dear reviewer,
the detailled point-by-point response can be seen in the attached word-file.
Sincerely

Reviewer 2 Report
The effect of lower body anaerobic pre-loading on upper body ergometer time trial performance
This is an interesting study about pre-exercise metabolic conditioning, the basic idea being: ‘to force an athlete’s metabolism towards aerobic metabolism by inhibiting anaerobic glycolysis during the first minute of exercise can preserve muscles abilities to perform on a high level for finishing the distance without losing overall performance. ‘ This is practically highly relevant, but my major problem is (see below) that the authors did not explain well enough why a 25 all-out exercise would be necessary to accomplish this.
Otherwise, the manuscript is reasonably well -written and the results are clearly presented. However, in addition to my major concerns I also have quite a few specific comments and questions.
In order of the manuscript these are :(I indicated which one are important)
Abstract: how much time between the precondition exercise and the time trial?( leaving approximately 9 min for recovery I read later on)
- Maximal double-poling performance after the TTlow was not significantly (p>0.05) faster 20 by 0.9±4.6 s, compared to the TThigh (225.1±17.6 s v. 226.1±15.7 s).
Suggest to rephrase: Maximal double-poling performance after the TTlow (225.1±17.6 s) was similar (p>0.05) to the TThigh (226.1±15.7 s).
- Net La increase from rest to the end of the TTlow was 10.5±2.2 mmol. l-1 and 6.5±3.4 mmol.l in TThigh (p<0.05). (this seems a rather obvious result, necessary to put this in the abstract?
-VCO2 was different (p<0.05) during : suggest put in the direction of the difference, higher /lower ?in TThigh
Importantly: I have some difficulties with the conclusion:
- anaerobic metabolic pre-conditioning leg exercise significantly inhibited net-La increase…..I am not sure you can state that it inhibited the increase, this suggest a causal relation, but when one measures blood lactate , this is the nett balance between production and disappearance, suggest to change to what was truly found: blood lactate increase was lower after TThigh
-I do not think the second part of the conclusion is right:
but all-out upper body performance was just slightly reduced. It wasn’t reduced since times were very similar (225 -226s). This has to be changed
-The pre-conditioning method may have some potential but needs to be combined with a pacing strategy different from the usual warm-up procedure. This remark comes a bit as a surprise, since there is no data about this in the abstract. Pacing? Wasn’t the 25 sprint all-out? Do you have indications that the power values were lower compared to when the athletes would do the same 25 s without a TT? So it raises all kinds of questions: if it is necessary to mention pacing than the authors should also present results on pacing
Introduction:
‘’e negatively affected by an intense warm up that decreases the availability of high-energy phosphates [9]. On the other hand, exercise longer than 2-3 min is primarily limited by aerobic energy contribution which may be enhanced by intense warm-up procedures [10].’
For his very reason a couple of short-lasting all-out sprints usually are incorporated in wup to speed up VO2 kinetics without decreasing PCr stores or pH to much. This is why I find the choice of a fatiguing 25 s maximal exercise so surprising. I assume that the authors will present the logic for this.
‘Although excessively intense exercise before a high-intensity bout may reduce performance level through decreased muscle glycogenolysis and La production [13–14], the exact role of lower levels of muscle glycogen and pH remain unclear’. ? Well, I think that it is rather well known that both are detrimental for performance
Lines 65-79: Pleas add that some of these findings may be more coincidental rather that a cause and effect.
In recent studies it has been found that a previous rise of blood La level decreased La production in a following maximal performance trial [21]. Similar results were found in studies of Müller et al. [4], Purge et al. [5] and Birnbaumer et al. [6]. Thus, priming effects can be reached by a short high-intensity exercise bout applying non-specific muscle groups and so speeding up VO2 kinetics highly important for a prolonged race duration [2,10].
It seems important to add (this may be obvious to the authors but not to me) that in the cited studies performance was similar or improved, otherwise, a lower lactate production after priming is meaningless.
Rephrase the following: One main problem in high-intensity prolonged exercise is to tolerate the side effects of the highly anaerobic first 40-60 s of exercise
Importantly: Please mention with all studies ( e.g. Purge et al. [5] line 87) with what muscle group the priming and time trials were performed, I got the impression that sometimes the priming and performance trial have been done with the same muscle group.. This is important throughout the introduction because the main question of this study relates to leg priming followed by arm TT.
Materials and methods
Please indicate clearly that the study approved by the (medical) ethics committee?….to sign written consent approved by the Medical Ethics Committee of the local university is a bit of a vague phrase
Page 4 lines 65: please also state here that the HIE was maximal (all out), I see it was in the legend of figure 1
Results
Important: Please only use faster and slower when significant. (throughout the manuscript) eg. P5 l191
In TTlow the subjects were slower in the first 400-m, but faster in the last 400-m (Table 2)….arguably they are nt slower in the fist 400m p>0.05 and small effect size. However, one could say that they tended to be faster (p= 0.07 and moderate effect sizes) in the last 400m
Table 3 type error: Effect zise:
Why was glucose measured? why would blood glucose be different between TTs ?
Importantly: If nett lactate increase in TThigh was lower (which only seems logical since they started with higher level prior to the TT), how positive can one be that this means that production by the arm muscles was lower? It seems that this is what is suggested by the authors (may be more lactate produced by the arms was taken up by the primed leg muscles or the heart (which very much likes to use lactate as a substrate because of a different LDH isoform)). I would phrase this more carefully in all sections of the manuscript (see my remark on the abstract)
Figure 4 is interesting. Delta200m=TTlow-TThigh, thus a positive value indicates that subjects who started out slower following TThigh compared to TTlow had faster total times, while subjects that did not adapt their pacing following TThigh or even went out slightly faster in the first 200m, showed a reduction of performance. (I think that a sentence like this one would be helpful: easier to read)
Discussion
The main finding of the present study was that 4 min maximal upper body performance time was not significantly influenced by high-intensity lower body cycling pre-load exercise although anaerobic energy contribution (of the arms is the suggestion) was substantially reduced (Table 2).
The authors are convinced that their lactate measures are a reflection of production…againI would phrase this more prudent….
MAJOR1:I think that the work would benefit by making clear why a 25 s all-out test was used. If it was just to increase VO2 kinetics at the start of the time trial, this doesn’t seem to be the smartest strategy, as the priming exercise is fatiguing which may negatively affect subsequent performance (the effect could even be psychological). A few shorter all-out sprints may be better.
MAJOR2: If the goal was to increase lactate, then the authors should explain carefully why they would expect blood lactate levels to have a positive effect on subsequent performance with the arms…there is one reference included suggesting that blood lactate would directly increase mitochondrial respiration. However, with this kind of short lasting exercise glycogen (and glucose) levels are not rate limiting, or are they? Mitochondrial respiration would largely depend on the ATP/ADP ratio and not on blood lactate. Why would the mitochondria start using more oxygen by feeding them with blood lactate (which has to be taken up by the arm muscle cells and converted via LDH into pyruvate, the substrate for mitochondrial respiration). The arm muscle fibres are loaded with glycogen they may not need additional lactate from the blood.
Thus, clearly presenting the EXACT rational for the choice of the present protocol would rely be more helpful, than the many effects of priming exercise that are no presented in a rather incoherent manner in the introduction. Especially since the reader may not be familiar with reference 4 and statements like ‘La levels by non-specific muscle exercise inhibited subsequent net La production, an effect which is suggested to be applied in competitive high-intensity exercise to improve performance [4].’ Are not very helpful (not specific enough)
Lines 254-259: this may all be true, but what is the intention of mentioning this? Please be explicit, about how you want to use these references in relation to your findings
L261 Importantly, the effect of a high-intensity anaerobic start needs to be elucidated, what do you mean? What start when? In general, in competition?
Line 307 Sprint exercise, which induced a lasting lactic acidosis, before a heavy exercise bout may be compared to HIE as the influence on VO2 response was similar [15]. I am not sure what exactly the authors mean here ‘may be compared’ do you mean that the HIE was sort of a maximal sprint? (yeas it was I think)
Lines 315-326, I think the authors try to read too much in their findings. May be the conclusion should just be: a 25 s maximal priming sprint with the lower body followed by 9 min of rest neither has a positive nor negative effect on upper body short lasting time trial performance. There may be other combinations of priming exercise duration and rest that will enhance performance, but in the present study only one combination was investigated (which again makes it so important to carefully explain why exactly the presented protocol was used)
Lines 327-334 are important
Conclusion: I think this conclusion is better, than the one presented in the abstract.
Author Response
Dear reviewer,
attached is the point-by-point response of our revision of the manuscript.
Sincerely

Round 2
Reviewer 1 Report
Dear Authors:
After read carefully the second version, we appreciate your effort to include all coments
Little final details are proposed
1) Table 1: MASS, Include BODY MASS
2) Line 122: 7 th, th in superindex
3) Line 270 ; p<0.05
4) Please correct carefully trought the text: blood lactate, blood lactate coventrstion (abreviature, etc)
Author Response
We thank the reviewer for the careful suggestions and revised all requests accordingly. The changes can be seen by track changes in the submitted and revised manuscript.
1) Table 1: MASS, Include BODY MASS - changed
2) Line 122: 7 th, th in superindex - changed
3) Line 270 ; p<0.05 - adapted
4) Please correct carefully trought the text: blood lactate, blood lactate coventrstion (abreviature, etc) - controlled and adapted where appropriate
Reviewer 2 Report
I have no further comments
Author Response
We thank the reviewer for the most valuable review process which clearly improved the manuscript.